# Innovative approaches in QSPR modelling using topological indices for the development of cancer treatments

Xiaolong Shi[1], Saeed Kosari[1]*, Masoud Ghods[2], Negar Kheirkhahan[2]

1 Institute of Computing Science and Technology, Guangzhou University, Guangzhou, China, 2 Department of Applied Mathematics, Semnan University, Semnan, Iran

* saeedkosari38@gzhu.edu.cn

**Data Availability Statement:** The data that support the findings of this study are openly available in [ChemSpider] at [http://www.chemspider.com/About Us.aspx]. Data are contained within the article.

## Abstract

This paper provides a comprehensive review of quantitative structure-property relationships (QSPR) about to cancer drugs, with a focus on the application of topological indices (TI) and data analysis techniques. Cancer is a serious and life-threatening disease for which no complete cure currently exists. Consequently, extensive research is ongoing to develop new therapeutic agents. The application of topological indices in chemistry and medicine, particularly in the investigation of the molecular, pharmacological, and therapeutic properties of drugs, has become a significant tool. This article investigates the potential of Temperature indices in analyzing the physicochemical properties of drugs used for cancer treatment. The approach employs QSPR modeling to establish correlations between the molecular structure of a compound and its physical and chemical properties. The analysis covers a range of Cancer drugs, including Aminopterin, Convolutamide A, Convolutamydine A, Daunorubicin, Minocycline, Podophyllotoxin, Caulibugulone E, Perfragilin A, Melatonin, Tambjamine K, Amathaspiramide E, and Aspidostomide E. The findings demonstrate that optimal regression models (Fifty-eight models) incorporating TI can effectively predict physicochemical properties, such as Boiling Point (BP), Enthalpy (EN), Flash Point (FP), Molar Refractivity (MR), Polar Surface Area (PSA), Surface Tension (ST), Molecular Volume (MV), and Complexity (COM). This research suggests that temperature-based topological indices (TI) are promising tools for the development and optimization of cancer drugs, as demonstrated by statistically significant results with a p-value less than 0.05. In addition to the linear regression model, which performed the best, two other machine learning models, namely SVR and Random Forest, were also used for further analysis and comparison of their performance in predicting the physicochemical properties of drugs, to assess the advantages and disadvantages of each model.

## 1 Introduction

In the treatment of this disease within the human body, alkylating agents and metabolites are commonly employed. Although significant attention is devoted to the development and

**Funding:** This work was supported by the National Natural Science Foundation of China under grants 62332006 and 62172302, with Xiaolong Shi as the principal recipient.

**Competing interests:** The authors have declared that no competing interests exist.

research of initial cancer therapies, the process of drug discovery, from identifying novel chemical compounds to obtaining regulatory approval, remains complex, costly, and time-intensive. Traditional approaches frequently encounter obstacles in compound synthesis and biological screening, leading the scientific community to explore more efficient methods for compound discovery. Chemical graph theory, an interdisciplinary field, is utilized to examine molecular structures and to establish correlations between activities, properties, and various phenomena. In this context, a molecular graph represents the structural formula of a chemical compound, with vertices corresponding to atoms and edges to chemical bonds. Chemical graph theory provides innovative tools for analyzing chemical structures, including topological indices, which serve as descriptors for the structure and specific properties of molecular graphs, typically represented as real numbers [1, 2]. Numerous studies have applied topological indices in the analysis of molecular graphs and drug structures [3–8]. A fundamental approach to exploring the relationship between a substance's physicochemical properties and its topological indices is through Quantitative Structure-Property Relationship (QSPR) models. These models use regression analysis to examine the correlations between physical and chemical properties and topological indices. Additionally, many studies in Quantitative Structure-Activity Relationship (QSAR) have applied topological indices to drug structures [9, 10]. In this article, various temperature-based indices are evaluated across several Cancer drugs, enabling researchers to identify the associated physical properties and chemical reactions. Furthermore, in addition to linear regression, we employed Support Vector Regression (SVR) and Random Forest models to explore and assess the predictive capabilities of these methods in determining the physicochemical properties of cancer drugs. These models were applied to identify the most effective model for predicting the properties of the drugs. The results of our analysis help in selecting the best predictive model, which is crucial for improving drug design and optimizing the therapeutic effectiveness of cancer treatments [11, 12].

In this study, the drug's structure is modeled as a graph where each vertex V(G) represents an atom and each edge E(G) signifies a chemical bond between atoms. The graphs considered are simple and connected. The degree of a vertex, defined as the number of edges incident to it, characterizes its connectivity [13].

## 2 Methodology and analysis

In this study, cancer drugs are modeled as simple graphs. To calculate the topological indices of these drug structures, we utilize techniques such as vertex partitioning, edge partitioning, and various computational methods. Our analysis is restricted to finite, simple, connected graphs. Let G denote a graph with a vertex set V and an edge set E. The degree $d_u$ of a vertex u is defined as the number of vertices adjacent to u. Below is a list of the topological formulas used in this study.

**Definition 2.1** Fajtlowicz defined the concept of vertex temperature u for a connected graph G as follows [14]:

$$T_u = \frac{d_u}{n - d_u} \tag{1}$$

**Definition 2.2** Product connectivity temperature index [15] is

$$PT(G) = \sum_{uv \in E(G)} \frac{1}{\sqrt{T_u \times T_v}} \tag{2}$$

**Definition 2.3** Harmonic temperature index [3] is

$$HT(G) = \sum_{uv \in E(G)} \frac{2}{T_u + T_v} \tag{3}$$

**Definition 2.4** Symmetric division temperature index [16] is

$$SDT(G) = \sum_{uv \in E(G)} \left( \frac{T_u}{T_v} + \frac{T_v}{T_u} \right) \tag{4}$$

**Definition 2.5** Modified third temperature index [17] is

$$^mT_3(G) = \sum_{uv \in E(G)} \frac{1}{(T_u + T_v)} \tag{5}$$

**Definition 2.6** Modified second temperature index [17] is

$$^mT_2(G) = \sum_{uv \in E(G)} \frac{1}{(T_u . T_v)} \tag{6}$$

**Definition 2.7** Second hyper temperature indices [2] is

$$HT_2(G) = \sum_{uv \in E(G)} (T_u \times T_v)^2 \tag{7}$$

**Definition 2.8** Sum connectivity temperature index [16] is

$$ST(G) = \sum_{uv \in E(G)} \frac{1}{\sqrt{T_u + T_v}} \tag{8}$$

**Definition 2.9** F-temperature index [16] is

$$FT(G) = \sum_{uv \in E(G)} (T_u^2 + T_v^2) \tag{9}$$

**Definition 2.10** Second temperature index [16] is

$$T_2(G) = \sum_{uv \in E(G)} (T_u \times T_v) \tag{10}$$

**Definition 2.11** Reciprocal product connectivity index [16] is

$$RPT(G) = \sum_{uv \in E(G)} \sqrt{(T_u \times T_v)} \tag{11}$$

**Definition 2.12** First hyper temperature indices [2] is

$$HT_1(G) = \sum_{uv \in E(G)} (T_u + T_v)^2 \tag{12}$$

A list of abbreviations used in the article is given in Table 1.

In recent years, scientists have increasingly utilized the QSPR/QSAR methodology to predict the physicochemical properties of chemical compounds through topological indices. This approach has been extensively applied in numerous studies to analyze a diverse array of drugs, including highly resistant anticancer agents, anti-COVID-19 drugs targeting the Omicron

**Table 1. Abbreviations list.**

| Meaning | Abbreviation |
|---|---|
| Boiling point | BP |
| Enthalpy of Vaporization | EN |
| Flash Point | FP |
| Molar Refractivity | MR |
| Polar Surface Area | PSA |
| Surface Tension | ST |
| Molar Volume | MV |
| Complexity | COM |
| Number of samples used for building the regression equation | N |
| Correlation coefficient | R |
| Standard error of regression coefficient | SE |
| Fisher's statistic | F |
| Topological indices | TI |
| Quantitative structure-property relationship | QSPR |

https://doi.org/10.6084/m9.figshare.26984122.v1

variant, breast cancer therapies, entropy tests involving benzene derivatives, nanotubes, Lyme disease treatments, and research on temperature indicators [18–22].

# 3 Mathematical computations of topological indices

This section presents the topological indices (TI) of cancer drugs and the QSPR modeling of their molecular structures.

## 3.1 Topological Index computation

Let A be a graph representing Aspidostomide E, where the edges are partitioned into distinct subsets based on specific criteria.

$$E_1 = \left\{ uv \in E(A) | T_u = \frac{1}{24}, T_v = \frac{2}{23} \right\}, E_2 = \left\{ uv \in E(A) | T_u = \frac{1}{24}, T_v = \frac{3}{22} \right\}, E_3 = \left\{ uv \in E(A) | T_u = \frac{2}{23}, T_v = \frac{2}{23} \right\},$$

$$E_4 = \left\{ uv \in E(A) | T_u = \frac{2}{23}, T_v = \frac{3}{22} \right\}, E_5 = \left\{ uv \in E(A) | T_u = \frac{3}{22}, T_v = \frac{3}{22} \right\}.$$

The study of the edges in A is shown in Table 2.

**Table 2. Dividing the edges of graph A.**

| $(\mathbf{T_u}, \mathbf{T_v})$ $\mathbf{uv} \in \mathbf{E(A)}$ | $\left(\frac{1}{24}, \frac{2}{23}\right)$ | $\left(\frac{1}{24}, \frac{3}{22}\right)$ | $\left(\frac{2}{23}, \frac{2}{23}\right)$ | $\left(\frac{2}{23}, \frac{3}{22}\right)$ | $\left(\frac{3}{22}, \frac{3}{22}\right)$ |
|---|---|---|---|---|---|
| **Number of edges** | 1 | 5 | 3 | 7 | 12 |

https://doi.org/10.6084/m9.figshare.26984170.v1

By applying Definitions 2.1 through 2.12, we obtain the following results:

$$1.\ PT(A) = \sum_{uv \in E(A)} \frac{1}{\sqrt{T_u \cdot T_v}} = \left(\frac{1}{\sqrt{\frac{1}{24} \cdot \frac{2}{23}}}\right) + 5\left(\frac{1}{\sqrt{\frac{1}{24} \cdot \frac{3}{22}}}\right) + 3\left(\frac{1}{\sqrt{\frac{2}{23} \cdot \frac{2}{23}}}\right) + 7\left(\frac{1}{\sqrt{\frac{2}{23} \cdot \frac{3}{22}}}\right) + 12\left(\frac{1}{\sqrt{\frac{3}{22} \cdot \frac{3}{22}}}\right) = 269.73$$

$$2.\ HT(A) = \sum_{uv \in E(A)} \frac{2}{T_u + T_v} = \left(\frac{2}{\frac{1}{24} + \frac{2}{23}}\right) + 5\left(\frac{2}{\frac{1}{24} + \frac{3}{22}}\right) + 3\left(\frac{2}{\frac{2}{23} + \frac{2}{23}}\right) + 7\left(\frac{2}{\frac{2}{23} + \frac{3}{22}}\right) + 12\left(\frac{2}{\frac{3}{22} + \frac{3}{22}}\right) = 256.91$$

$$3.\ SDT(A) = \sum_{uv \in E(A)} \left(\frac{T_u}{T_v} + \frac{T_v}{T_u}\right) = \left(\frac{\frac{1}{24}}{\frac{2}{23}} + \frac{\frac{2}{23}}{\frac{1}{24}}\right) + 5\left(\frac{\frac{1}{24}}{\frac{3}{22}} + \frac{\frac{3}{22}}{\frac{1}{24}}\right) + 3\left(\frac{\frac{2}{23}}{\frac{2}{23}} + \frac{\frac{2}{23}}{\frac{2}{23}}\right) + 7\left(\frac{\frac{2}{23}}{\frac{3}{22}} + \frac{\frac{3}{22}}{\frac{2}{23}}\right) + 12\left(\frac{\frac{3}{22}}{\frac{3}{22}} + \frac{\frac{3}{22}}{\frac{3}{22}}\right) = 65.8990$$

$$4.\ Tm_3(A) = \sum_{uv \in E(A)} \frac{1}{(T_u + T_v)} = \left(\frac{1}{\frac{1}{24} + \frac{2}{23}}\right) + 5\left(\frac{1}{\frac{1}{24} + \frac{3}{22}}\right) + 3\left(\frac{1}{\frac{2}{23} + \frac{2}{23}}\right) + 7\left(\frac{1}{\frac{2}{23} + \frac{3}{22}}\right) + 12\left(\frac{1}{\frac{3}{22} + \frac{3}{22}}\right) = 128.4550$$

$$5.\ Tm_2(A) = \sum_{uv \in E(A)} \frac{1}{(T_u \cdot T_v)} = \left(\frac{1}{\frac{1}{24} \cdot \frac{2}{23}}\right) + 5\left(\frac{1}{\frac{1}{24} \cdot \frac{3}{22}}\right) + 3\left(\frac{1}{\frac{2}{23} \cdot \frac{2}{23}}\right) + 7\left(\frac{1}{\frac{2}{23} \cdot \frac{3}{22}}\right) + 12\left(\frac{1}{\frac{3}{22} \cdot \frac{3}{22}}\right) = 2788.4170$$

$$6.\ HT_2(A) = \sum_{uv \in E(A)} (T_u \times T_v)^2 = \left(\frac{1}{24} \cdot \frac{2}{23}\right)^2 + 5\left(\frac{1}{24} \cdot \frac{3}{22}\right)^2 + 3\left(\frac{2}{23} \cdot \frac{2}{23}\right)^2 + 7\left(\frac{2}{23} \cdot \frac{3}{22}\right)^2 + 12\left(\frac{3}{22} \cdot \frac{3}{22}\right)^2 = 0.0055$$

$$7.\ FT(A) = \sum_{uv \in E(A)} (T_u^2 + T_v^2) = \left(\left(\frac{1}{24}\right)^2 + \left(\frac{2}{23}\right)^2\right) + 5\left(\left(\frac{1}{24}\right)^2 + \left(\frac{3}{22}\right)^2\right) + 3\left(\left(\frac{2}{23}\right)^2 + \left(\frac{2}{23}\right)^2\right) + 7\left(\left(\frac{2}{23}\right)^2 + \left(\frac{2}{22}\right)^2\right) + 12\left(\left(\frac{3}{22}\right)^2 + \left(\frac{3}{22}\right)^2\right) = 0.7857$$

$$8.\ T_2(A) = \sum_{uv \in E(A)} (T_u \times T_v) = \left(\frac{1}{24} \cdot \frac{2}{23}\right) + 5\left(\frac{1}{24} \cdot \frac{3}{22}\right) + 3\left(\frac{2}{23} \cdot \frac{2}{23}\right) + 7\left(\frac{2}{23} \cdot \frac{3}{22}\right) + 12\left(\frac{3}{22} \cdot \frac{3}{22}\right) = 0.3609$$

$$9.\ RPT(A) = \sum_{uv \in E(A)} \sqrt{(T_u \times T_v)} = \sqrt{\frac{1}{24} \cdot \frac{2}{23}} + 5\sqrt{\frac{1}{24} \cdot \frac{3}{22}} + 3\sqrt{\frac{2}{23} \cdot \frac{2}{23}} + 7\sqrt{\frac{2}{23} \cdot \frac{3}{22}} + 12\sqrt{\frac{3}{22} \cdot \frac{3}{22}} = 3.0966$$

$$10.\ HT_1(A) = \sum_{uv \in E(A)} (T_u + T_v)^2 = \left(\frac{1}{24} + \frac{2}{23}\right)^2 + 5\left(\frac{1}{24} + \frac{3}{22}\right)^2 + 3\left(\frac{2}{23} + \frac{2}{23}\right)^2 + 7\left(\frac{2}{23} + \frac{3}{22}\right)^2 + 12\left(\frac{3}{22} + \frac{3}{22}\right)^2 = 1.5074$$

$$11.\ ST(A) = \sum_{uv \in E(A)} \frac{1}{\sqrt{T_u + T_v}} = \left(\frac{1}{\sqrt{\frac{1}{24} + \frac{2}{23}}}\right) + 5\left(\frac{1}{\sqrt{\frac{1}{24} + \frac{3}{22}}}\right) + 3\left(\frac{1}{\sqrt{\frac{2}{23} + \frac{2}{23}}}\right) + 7\left(\frac{1}{\sqrt{\frac{2}{23} + \frac{3}{22}}}\right) + 12\left(\frac{1}{\sqrt{\frac{3}{22} + \frac{3}{22}}}\right) = 59.6230$$

Fig 1 shows the Chemical structure and Molecular graph of Aspidostomide E.

Topological indices for other drugs can be computed using the methods described in Eqs (1) to (12) from Section 2. The indices are detailed in Tables 3, 4, and Fig 2 illustrates the drugs. Additional information about these drugs can be accessed on Chemical book [23], and Table 5 summarizes their physical and chemical properties [15, 24].

## 3.2 Discussion and comparison of advanced machine learning models and linear models for QSPR analysis

The primary objective of this section is to conduct a QSPR analysis of various topological indices (TI) and examine their correlation with several physicochemical properties and activities of drugs. The drugs under investigation include Aminopterin, Convolutamide A, Convolutamydine A, Daunorubicin, Minocycline, Podophyllotoxin, Caulibugulone E, Perfragilin A, Melatonin, Tambjamine K, Amathaspiramide E, and Aspidostomide E. We assessed the effectiveness of these TI in predicting drug properties. We analyzed eight physicochemical properties: Boiling Point (BP), Enthalpy (EN), Flash Point (FP), Molar Refractivity (MR), Polar Surface Area (PSA), Surface Tension (ST), Molecular Volume (MV), and Complexity (COM),

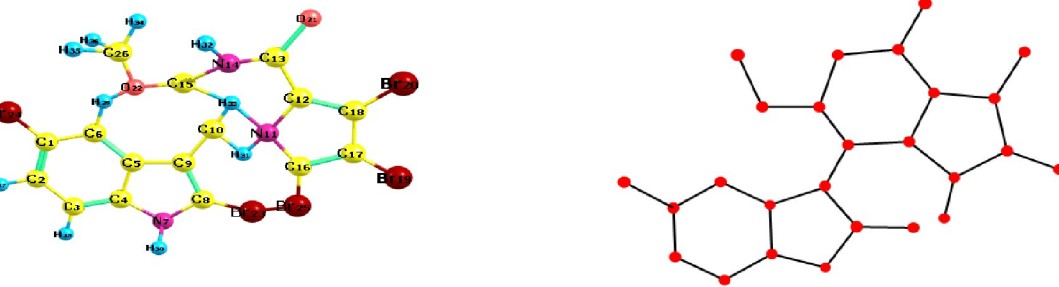

**Fig 1. Chemical structure and Molecular graph of Aspidostomide E.** a) Chemical structure of Aspidostomide E. b) Molecular graph of Aspidostomide E. https://doi.org/10.6084/m9.figshare.26984881.v1.

with values obtained from PubChem and Chemspider. Table 6 displays the correlation coefficients (r) between these physicochemical attributes and the degree-based topological indices. Tables 7–13 demonstrate that a linear QSPR model provides the best fit for predicting these properties. The values are normally distributed, and fifty-eight regression models were employed for data analysis. Notably, the PT(G), HT(G), $^m T_3(G)$, $T_2(G)$, and SDT(G) indices exhibit high correlations with COM, with R-values of 0.913, 0.905, 0.908, 0.915, and 0.905, respectively. Additionally, the ST) G (index shows a strong positive correlation with MR, with r = 0.924. In contrast, the RPT(G) topological index does not show a significant correlation with any physicochemical feature. The $HT_1(G)$ and $T_2(G)$ indices have a significant inverse correlation with MR and MV. The $HT_2(G)$ index is identified as the best predictor for BP, EN, MR, and MV, demonstrating an inverse correlation.

Advanced machine learning models, including **SVR, Random Forest**, and **Linear Regression** (a traditional model), were employed for the analysis. The findings revealed the following key observations:

- SVR and Linear Regression models exhibited superior performance in predicting physicochemical properties, achieving correlation coefficients (r) above 0.9 for most properties. These results underscore the high predictive power of advanced machine learning techniques in QSPR analysis (Vapnik, 1995; Seber & Lee, 2003).

**Table 3. The values of the temperature indices of the drugs.**

| Drugs | PT(G) | HT(G) | FT(G) | $^m T_3(G)$ | $^m T_2(G)$ | $T_2(G)$ |
|---|---|---|---|---|---|---|
| Aminopterin | 424.80 | 404.84 | .4937 | 202.4217 | 5638.8333 | .2174 |
| Convolutamide A | 334.18 | 326.50 | .4434 | 163.2518 | 4312.0000 | .0843 |
| Convolutamydine A | 113.89 | 103.34 | 1.3709 | 51.6685 | 768.5972 | .4191 |
| Daunorubicin | 635.63 | 597.36 | .4936 | 298.6796 | 10270.3333 | .2072 |
| Minocycline | 453.42 | 421.33 | .4965 | 210.6655 | 6729.2500 | .2572 |
| Podophyllotoxin | 242.93 | 232.44 | .6013 | 116.2185 | 3116.0000 | .2794 |
| Caulibugulone E | 78.76 | 60.41 | .8717 | 37.5380 | 445.1110 | .3654 |
| PerfragilinA | 116.41 | 107.78 | 1.2321 | 53.8880 | 820.7780 | .5396 |
| Melatonin | 120.81 | 115.78 | 1.0205 | 57.8890 | 852.8890 | .4552 |
| Tambjamine k | 154.23 | 155.50 | .5086 | 74.3290 | 1243.3890 | .2350 |
| Amathaspiramide E | 180.71 | 165.53 | .8283 | 85.2550 | 1590.0000 | .3553 |
| Aspidostomide E | 269.73 | 256.91 | .7857 | 128.4550 | 2788.4170 | .3609 |

https://doi.org/10.6084/m9.figshare.27002857.v1

**Table 4. The values of the temperature indices of the drugs.**

| Drugs | RPT(G) | SDT(G) | HT$_1$(G) | HT$_2$(G) | ST(G) |
|---|---|---|---|---|---|
| Aminopterin | 2.6411 | 78.7029 | .9286 | .0016 | 81.5373 |
| Convolutamide A | 1.2272 | 59.9719 | .3764 | .0004 | 76.5136 |
| Convolutamydine A | 2.5316 | 50.3386 | 1.9278 | .0129 | 30.3491 |
| Daunorubicin | 2.8443 | 103.8873 | .8853 | .0012 | 111.4924 |
| Minocycline | 2.9377 | 84.7168 | 1.0979 | .0023 | 88.9976 |
| Podophyllotoxin | 2.7251 | 47.0252 | 1.1602 | .0033 | 63.9580 |
| Caulibugulone E | 1.9689 | 35.5210 | 1.6025 | .0137 | 23.5640 |
| PerfragilinA | 3.0151 | 46.3500 | 2.3113 | .0198 | 30.9370 |
| Melatonin | 2.8059 | 42.1070 | 1.9309 | .0133 | 32.1270 |
| Tambjamine k | 2.0497 | 45.9230 | .9786 | .0031 | 38.3870 |
| Amathaspiramide E | 2.6621 | 54.1030 | 1.5389 | .0071 | 43.0260 |
| Aspidostomide E | 3.0966 | 65.8990 | 1.5074 | .0055 | 59.6230 |

https://doi.org/10.6084/m9.figshare.27002857.v1

- The Random Forest model also showed acceptable performance. Although its accuracy was slightly lower than that of the tuned SVR and Linear Regression models, it provided valuable insights into the relationships between topological indices and drug properties (Breiman, 2001).

- In contrast, the SVR model demonstrated weaker performance, with lower correlation coefficients, highlighting the necessity of parameter optimization for achieving accurate predictions (Vapnik, 1995).

Fig 3 provides a graphical representation of the correlations between TI and physicochemical properties. Fig 4 illustrates the relationship between TI and the physical properties of the drugs studied.

### 3.3 QSPR analysis

Building upon the temperature indices computed in Section 2, this section aims to develop a linear regression model. This model will be used to elucidate the relationships between the

**Table 5. Physicochemical properties of cancer drug.**

| Drugs | BP | EN | FP | MR | PSA | ST | MV | COM |
|---|---|---|---|---|---|---|---|---|
| Aminopterin | - | - | - | 114.3 | 219 | 103.3 | 277.2 | 674 |
| Convolutamide A | 629.9 | 97.9 | 334.7 | 130.1 | 78 | 51.9 | 396 | - |
| Convolutamydine A | 504.9 | 81.6 | 259.2 | 68.2 | 66 | 59 | 190 | 363 |
| Daunorubicin | 770 | 117.6 | 419.5 | 130 | 186 | 87.4 | 339.4 | 960 |
| Minocycline | 803.3 | 122.5 | 439.6 | 116 | 165 | 90 | 294.6 | 971 |
| Podophyllotoxin | 597.9 | 93.6 | 210.2 | 104.3 | 93 | 52.8 | 302.4 | 583 |
| Caulibugulone E | 373 | 62 | 179.4 | 52.2 | 66 | 53.2 | 139.1 | 319 |
| PerfragilinA | 431.5 | 68.7 | 214.8 | 63.6 | 106 | 68.9 | 167.8 | 543 |
| Melatonin | 512.8 | 78.4 | 264 | 67.6 | 54 | 46.7 | 197.6 | 301 |
| Tambjamine k | 391.7 | 64.1 | 190.7 | 76.6 | 49 | 37 | 235.1 | 293 |
| Amathaspiramide E | 572.7 | 90.3 | 300.2 | 89.4 | 62 | 56.2 | 233.9 | 489 |
| Aspidostomide E | 798.8 | 116.2 | 436.9 | 116 | 59 | 71.4 | 262 | 572 |

https://doi.org/10.6084/m9.figshare.26985559.v1

### caulibugulone E

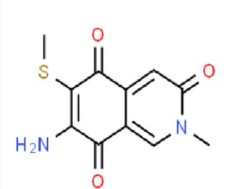

| | |
|---|---|
| Molecular Formula | $C_{10}H_9N_3O$ |
| Average mass | 187.198 Da |
| Monoisotopic mass | 187.074554 Da |
| ChemSpider ID | 10479941 |

### perfragilin A

| | |
|---|---|
| Molecular Formula | $C_{11}H_{10}N_2O_3S$ |
| Average mass | 250.274 Da |
| Monoisotopic mass | 250.041214 Da |
| ChemSpider ID | 140262 |

### Melatonin

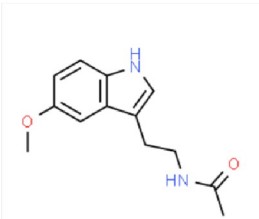

| | |
|---|---|
| Molecular Formula | $C_{13}H_{16}N_2O_2$ |
| Average mass | 232.278 Da |
| Monoisotopic mass | 232.121185 Da |
| ChemSpider ID | 872 |

### tambjamine K

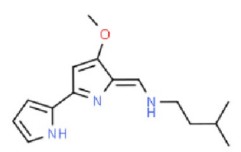

| | |
|---|---|
| Molecular Formula | $C_{15}H_{21}N_3O$ |
| Average mass | 259.347 Da |
| Monoisotopic mass | 259.168457 Da |
| ChemSpider ID | 24676448 |

— Double-bond stereo

### Amathaspiramide E

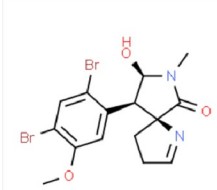

| | |
|---|---|
| Molecular Formula | $C_{15}H_{16}Br_2N_2O_3$ |
| Average mass | 432.107 Da |
| Monoisotopic mass | 429.952759 Da |
| ChemSpider ID | 8653301 |

— 3 of 3 defined stereocentres

### (3R,4S)-6,7,8-Tribromo-4-(2,5-dibromo-1H-indol-3-yl)-3-methoxy-3,4-dihydropyrrolo[1,2-a]pyrazin-1(2H)-one

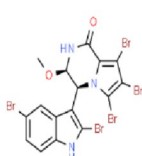

| | |
|---|---|
| Molecular Formula | $C_{16}H_{10}Br_5N_3O_2$ |
| Average mass | 675.789 Da |
| Monoisotopic mass | 670.668945 Da |
| ChemSpider ID | 32674911 |

### Aminopterin

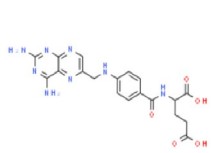

| | |
|---|---|
| Molecular Formula | $C_{19}H_{20}N_8O_5$ |
| Average mass | 440.413 Da |
| Monoisotopic mass | 440.155670 Da |
| ChemSpider ID | 2069 |

### 3-(3,4-Dibromo-5-hydroxyphenyl)-3-hydroxy-1-tetradecanoyl-2-pyrrolidinone

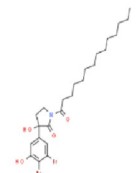

| | |
|---|---|
| Molecular Formula | $C_{24}H_{35}Br_2NO_4$ |
| Average mass | 561.347 Da |
| Monoisotopic mass | 559.093262 Da |
| ChemSpider ID | 8523269 |

### convolutamydine A

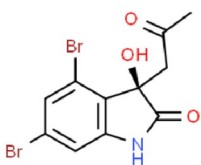

| | |
|---|---|
| Molecular Formula | $C_{11}H_9Br_2NO_3$ |
| Average mass | 363.002 Da |
| Monoisotopic mass | 360.894897 Da |
| ChemSpider ID | 4482165 |

— 1 of 1 defined stereocentres

### Daunorubicin

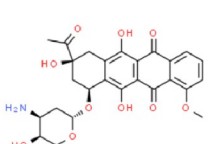

| | |
|---|---|
| Molecular Formula | $C_{27}H_{29}NO_{10}$ |
| Average mass | 527.520 Da |
| Monoisotopic mass | 527.179138 Da |
| ChemSpider ID | 28163 |

— 6 of 6 defined stereocentres

### Minocycline

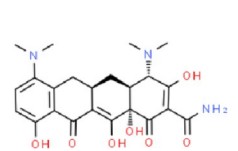

| | |
|---|---|
| Molecular Formula | $C_{23}H_{27}N_3O_7$ |
| Average mass | 457.476 Da |
| Monoisotopic mass | 457.184906 Da |
| ChemSpider ID | 16735907 |

— 4 of 4 defined stereocentres

### (-)-Podophyllotoxin

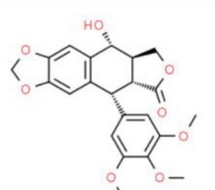

| | |
|---|---|
| Molecular Formula | $C_{22}H_{22}O_8$ |
| Average mass | 414.405 Da |
| Monoisotopic mass | 414.131470 Da |
| ChemSpider ID | 10162 |

— 4 of 4 defined stereocentres

**Fig 2. Chemical structure of cancer drugs from ChemSpider.** https://doi.org/10.6084/m9.figshare.26984215.v1.

**Table 6. Correlation coefficients of physical properties of drugs.**

| Drugs | BP | EN | FP | MR | PSA | ST | MV | COM |
|---|---|---|---|---|---|---|---|---|
| PT(G) | 0.816 | 0.839 | 0.784 | 0.87 | 0.807 | 0.735 | 0.768 | 0.913 |
| FT(G) | -0.467 | -0.484 | -0.385 | -0.742 | -0.444 | -0.262 | -0.765 | -0.54 |
| HT(G) | 0.815 | 0.837 | 0.781 | 0.882 | 0.798 | 0.721 | 0.787 | 0.905 |
| $^m T_3$(G) | 0.815 | 0.838 | 0.781 | 0.878 | 0.802 | 0.726 | 0.781 | 0.908 |
| $T_2$(G) | -0.408 | -0.444 | -0.355 | -0.758 | -0.365 | -0.178 | -0.85 | -0.506 |
| $^m T_2$(G) | 0.774 | 0.802 | 0.738 | 0.829 | **0.808** | 0.716 | 0.741 | 0.915 |
| RPT(G) | 0.369 | 0.341 | 0.318 | -0.005 | 0.291 | 0.471 | -0.253 | 0.574 |
| $HT_2$(G) | -0.626 | -0.65 | -0.531 | -0.854 | -0.413 | -0.292 | -0.863 | -0.512 |
| $HT_1$(G) | -0.424 | -0.455 | -0.362 | -0.772 | -0.368 | -0.178 | -0.859 | -0.512 |
| ST(G) | **0.836** | 0.86 | 0.767 | **0.924** | 0.764 | 0.682 | **0.848** | **0.921** |
| SDT(G) | 0.835 | **0.854** | **0.857** | 0.798 | 0.802 | **0.808** | 0.64 | 0.905 |

https://doi.org/10.6084/m9.figshare.26988049.v1

**Table 7. Statistical metrics for the linear QSPR model applied to PT (G).**

| Physical Properties | N | A | b | R | $R^2$ | SE | F | P | Indicator |
|---|---|---|---|---|---|---|---|---|---|
| BP | 11 | .748 | 396.850 | .816 | .666 | 52.158 | 17.908 | .000 | significant |
| EN | 11 | .106 | 64.216 | .839 | .705 | 6.753 | 21.467 | .000 | significant |
| FP | 11 | .453 | 184.053 | .784 | .615 | 35.291 | 14.360 | .001 | significant |
| MR | 12 | .141 | 57.416 | **.870** | **.756** | 7.768 | 31.028 | .000 | significant |
| PSA | 12 | .271 | 29.617 | .807 | .651 | 19.323 | 18.668 | .002 | significant |
| ST | 12 | .085 | 42.638 | .735 | .540 | 7.654 | 11.731 | .000 | significant |
| MV | 12 | .333 | 166.264 | .768 | .589 | 27.034 | 14.356 | .000 | significant |
| COM | 11 | 1.237 | 237.722 | .913 | .834 | 56.233 | 45.123 | .002 | significant |

The significance of bold numbers denote highest correlation value

https://doi.org/10.6084/m9.figshare.26990038.v1

**Table 8. Statistical metrics for the linear QSPR model applied to HT (G).**

| Physical Properties | N | A | b | R | $R^2$ | SE | F | P | Indicator |
|---|---|---|---|---|---|---|---|---|---|
| BP | 11 | .788 | 398.336 | .815 | .664 | 52.025 | 17.786 | .000 | significant |
| EN | 11 | .112 | 64.457 | .837 | .701 | 6.758 | 21.133 | .000 | significant |
| FP | 11 | .476 | 185.297 | .781 | .610 | 35.358 | 14.048 | .001 | significant |
| MR | 12 | .150 | 57.178 | **.882** | **.777** | 7.384 | 34.931 | .000 | significant |
| PSA | 12 | .282 | 30.878 | .798 | .637 | 19.599 | 17.577 | .002 | significant |
| ST | 12 | .088 | 43.220 | .721 | .519 | 7.781 | 10.809 | .000 | significant |
| MV | 12 | .359 | 164.758 | .787 | .619 | 25.904 | 16.252 | .000 | significant |
| COM | 11 | 1.295 | 243.005 | .905 | .820 | 58.209 | 40.922 | .002 | significant |

The significance of bold numbers denote highest correlation value

https://doi.org/10.6084/m9.figshare.26990662.v1

**Table 9. Statistical metrics for the linear QSPR model applied to SDT (G).**

| Physical Properties | N | A | b | R | $R^2$ | SE | F | P | Indicator |
|---|---|---|---|---|---|---|---|---|---|
| BP | 11 | 6.469 | 206.676 | .835 | .697 | 86.617 | 20.724 | .001 | significant |
| EN | 11 | .912 | 37.551 | .854 | .730 | 11.271 | 24.325 | .009 | significant |
| FP | 11 | 4.187 | 53.365 | **.857** | **.735** | 51.095 | 24.950 | .001 | significant |
| MR | 12 | 1.091 | 29.064 | .798 | .637 | 16.308 | 17.554 | .002 | significant |
| PSA | 12 | 2.279 | -35.475 | .802 | .643 | 33.620 | 18.029 | .002 | significant |
| ST | 12 | .792 | 17.651 | .808 | .653 | 11.432 | 18.829 | .001 | significant |
| MV | 12 | 2.345 | 113.318 | .640 | .409 | 55.790 | 6.927 | .025 | significant |
| COM | 11 | 10.271 | -59.578 | .905 | .819 | 101.305 | 40.625 | .000 | significant |

The significance of bold numbers denote highest correlation value

https://doi.org/10.6084/m9.figshare.26995264.v1

**Table 10. Statistical metrics for the linear QSPR model applied to $^m T_3$ (G).**

| Physical Properties | N | A | b | R | $R^2$ | SE | F | P | Indicator |
|---|---|---|---|---|---|---|---|---|---|
| BP | 11 | 1.589 | 395.950 | .815 | .664 | 52.542 | 17.751 | .000 | significant |
| EN | 11 | .225 | 64.102 | .838 | .702 | 6.816 | 21.176 | .000 | significant |
| FP | 11 | .961 | 183.747 | .781 | .610 | 35.651 | 14.094 | .001 | significant |
| MR | 12 | .301 | 56.854 | **.878** | **.772** | 7.547 | 33.795 | .000 | significant |
| PSA | 12 | .572 | 29.724 | .802 | .643 | 19.634 | 17.972 | .002 | significant |
| ST | 12 | .179 | 42.788 | .726 | .527 | 7.788 | 11.144 | .000 | significant |
| MV | 12 | .718 | 164.329 | .781 | .610 | 26.457 | 15.621 | .000 | significant |
| COM | 11 | 2.618 | 238.144 | .908 | .824 | 57.996 | 42.193 | .003 | significant |

The significance of bold numbers denote highest correlation value

https://doi.org/10.6084/m9.figshare.26995768.v1

**Table 11. Statistical metrics for the linear QSPR model applied to $^m T_2$ (G).**

| Physical Properties | N | A | b | R | $R^2$ | SE | F | P | Indicator |
|---|---|---|---|---|---|---|---|---|---|
| BP | 11 | .040 | 462.043 | .774 | .599 | 45.232 | 13.450 | .000 | significant |
| EN | 11 | .006 | 73.347 | .802 | .643 | 5.884 | 16.184 | .000 | significant |
| FP | 11 | .024 | 224.106 | .738 | .545 | 30.383 | 10.776 | .000 | significant |
| MR | 12 | .008 | 69.634 | **.829** | **.688** | 6.996 | 22.044 | .000 | significant |
| PSA | 12 | .015 | 50.813 | .808 | .654 | 15.325 | 18.874 | .008 | significant |
| ST | 12 | .005 | 49.727 | .716 | .512 | 6.273 | 10.494 | .000 | significant |
| MV | 12 | .018 | 194.512 | .741 | .549 | 22.557 | 12.162 | .000 | significant |
| COM | 11 | .070 | 333.617 | .915 | .838 | 44.485 | 46.431 | .000 | significant |

The significance of bold numbers denote highest correlation value

https://doi.org/10.6084/m9.figshare.26996149.v1

**Table 12. Statistical metrics for the linear QSPR model applied to $HT_2$ (G).**

| Physical Properties | N | A | b | R | $R^2$ | SE | F | P | Indicator |
|---|---|---|---|---|---|---|---|---|---|
| BP | 11 | -15380.93 | 696.092 | .626 | .392 | 61.777 | 5.804 | .000 | significant |
| EN | 11 | -2199.264 | 106.779 | .650 | .422 | 8.297 | 6.578 | .000 | significant |
| MR | 12 | -3735.217 | 120.232 | **.854** | **.730** | 6.662 | 27.046 | .000 | significant |
| MV | 12 | -10116.120 | 323.901 | .863 | .745 | 17.370 | 29.183 | .000 | significant |

The significance of bold numbers denote highest correlation value

https://doi.org/10.6084/m9.figshare.26996587.v1

temperature indices and the physical and chemical properties of the drugs.

$$P = B + A(TI) \tag{13}$$

Where:

- P: Represents the Anxiety drug property (dependent variable)

- B: Constant term (y-intercept)

- A: Regression coefficient

- TI: Topological index (independent variable)

Eq (13) represents the formulated linear regression model. In this equation, "P" denotes a specific property of an anxiety drug that we aim to predict or analyze. "B" is the constant term, and "A" is the regression coefficient, which indicates the change in "P" associated with a unit increase in the topological index. The analysis was performed using SPSS software to develop linear models for eight specific properties of cancer drugs across twelve different drugs. These models are based on the eleven topological indices computed earlier. The following section will present the various linear models tailored to each of the eight drug properties, using Eq (13) as the general framework.

### 3.4 Linear regression models

In this section, the linear regression models for topological indices (TI) are discussed using Eq (13). Tables 7–13 present the parameters and QSPR models associated with these TI. The following linear models for temperature indices are derived based on Eq (13):

**1. Product connectivity temperature index [PT (G)]**
BP = 396.850+0.748 [PT (G)], EN = 64.216+0.106 [PT (G)], FP = 184.053+0.453 [PT (G)]
MR = 57.416+0.141 [PT (G)], PSA = 29.617+0.271 [PT (G)], ST = 42.638+0.085 [PT (G)]
MV = 166.264+0.333 [PT (G)], COM = 237.722+1.237 [PT (G)]

**2. Harmonic temperature index [HT (G)]**
BP = 398.336+0.788 [HT (G)], EN = 64.457+0.112 [HT (G)], FP = 185.297+0.476 [HT (G)]

**Table 13. Statistical metrics for the linear QSPR model applied to FT (G).**

| Physical Properties | N | A | b | R | $R^2$ | SE | F | P | Indicator |
|---|---|---|---|---|---|---|---|---|---|
| MR | 12 | -65.545 | 143.984 | .742 | .550 | 15.348 | 12.239 | .000 | significant |
| MV | 12 | -181.194 | 391.033 | **.765** | **.585** | 39.540 | 14.093 | .000 | significant |

The significance of bold numbers denote highest correlation value

https://doi.org/10.6084/m9.figshare.26997142.v1

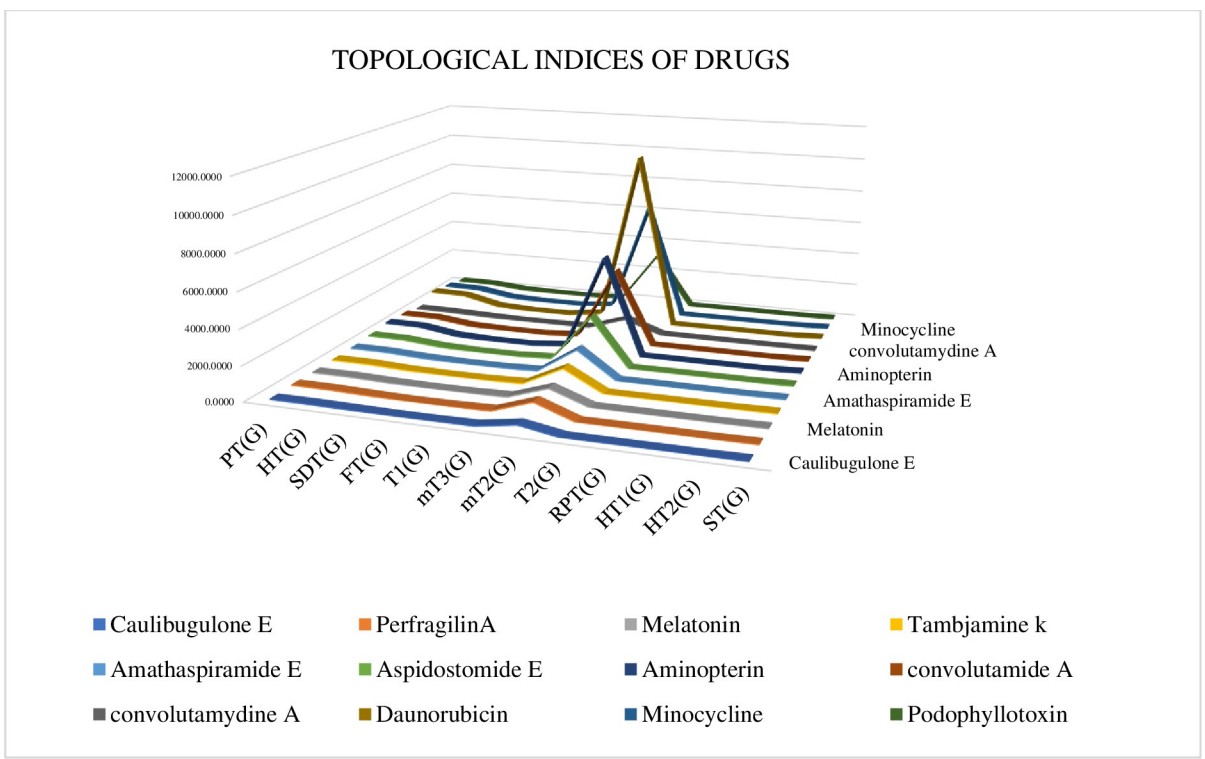

**Fig 3. Two-dimensional (2D) graph illustrating the relationship between drugs and their topological indices.** https://doi.org/10.6084/m9.figshare.26983915.v.

MR = 57.178+0.150 [HT (G)], PSA = 30.878+0.282 [HT (G)], ST = 43.220+0.088 [HT (G)]

MV = 164.758+0.359 [HT (G)], COM = 243.005+1.295 [HT (G)]

**3. Symmetric division temperature index [SDT (G)]**

BP = 186.910+6.469 [SDT (G)], EN = 37.003+.912 [SDT (G)], FP = 66.890+4.187 [SDT (G)]

MR = 26.265+1.091 [SDT (G)], PSA = -45232.412+2.279 [SDT (G)], ST = 10.415+0.792 [SDT (G)]

MV = 17.235+2.345 [SDT (G)], COM = 0.494+10.271 [SDT (G)]

**4. Modified third temperature index [$^{m}T_3$ (G)]**

BP = 395.950+1.589 [$^{m}T_3$ (G)], EN = 64.102+0.225 [$^{m}T_3$ (G)], FP = 183.747+0.961 [$^{m}T_3$ (G)]

MR = 56.854+0.301 [$^{m}T_3$ (G)], PSA = 29.724+0.572 [$^{m}T_3$ (G)], ST = 42.788+0.179 [$^{m}T_3$ (G)]

MV = 164.329+0.718 [$^{m}T_3$ (G)], COM = 238.144+2.618 [$^{m}T_3$ (G)]

**5. Modified second temperature index [$^{m}T_2$ (G)]**

BP = 462.043+0.040 [$^{m}T_2$ (G)], EN = 73.347+0.006 [$^{m}T_2$ (G)], FP = 224.106+0.024 [$^{m}T_2$ (G)]

MR = 69.634+0.008 [$^{m}T_2$ (G)], PSA = 50.813+0.015 [$^{m}T_2$ (G)], ST = 49.727+0.005 [$^{m}T_2$ (G)]

MV = 194.512+0.018 [$^{m}T_2$ (G)], COM = 333.617+0.070 [$^{m}T_2$ (G)]

**6. Second hyper temperature indices [$HT_2$ (G)]**

BP = 696.092–15380.93 [$HT_2$ (G)], EN = 106.779–2199.264 [$HT_2$ (G)]

MR = 120.232–3735.217 [$HT_2$ (G)], MV = 323.901–10116.120 [$HT_2$ (G)]

**7. F-temperature index FT (G)**

MR = 120.232–3735.217 [FT (G)], MV = 323.901–10116.120 [FT (G)]

**8. First hyper temperature indices $HT_1$ (G)**

MR = 120.232–3735.217 [$HT_1$ (G)], MV = 323.901–10116.120 [$HT_1$ (G)]

**9. Second temperature index $T_2$ (G)**

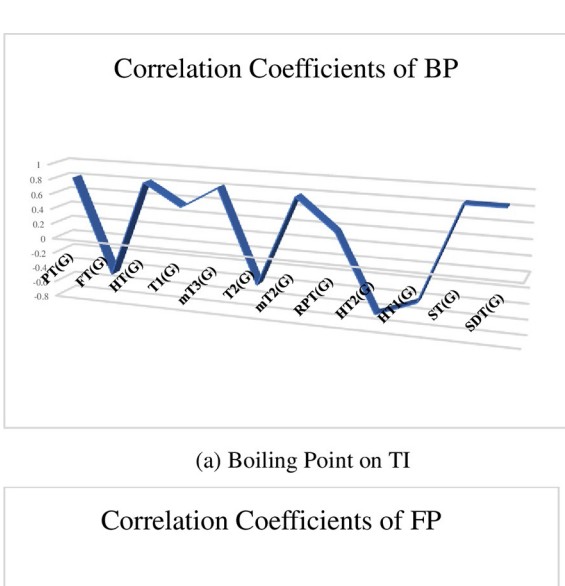

(a) Boiling Point on TI

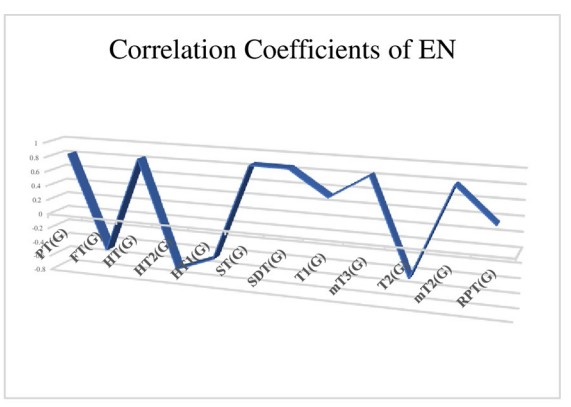

(b) Enthalpy on TI

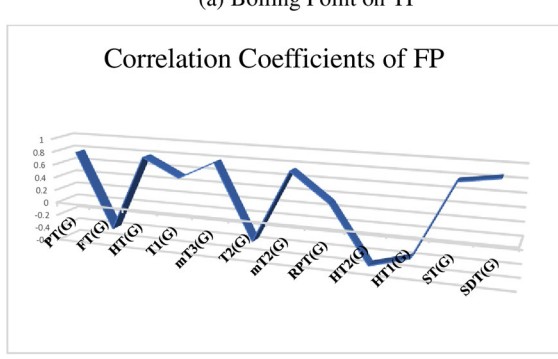

(c) Flash point on TI

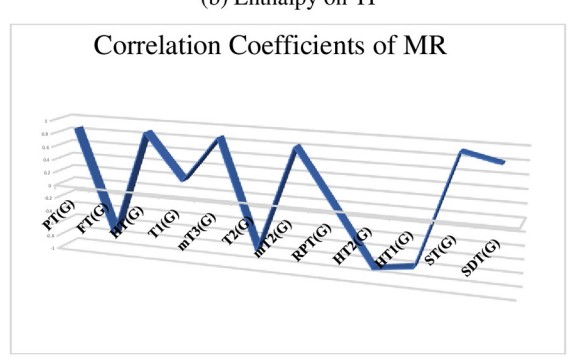

(d) Molar Refractivity on TI

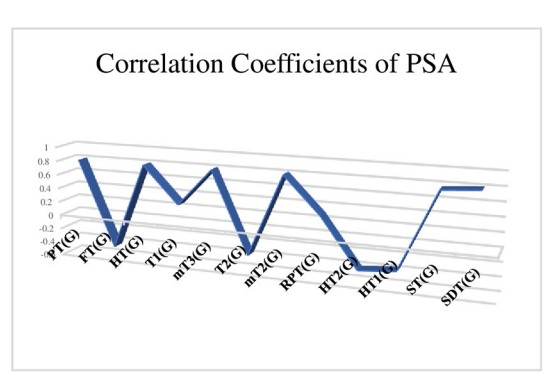

(e) Polar Surface Area on TI

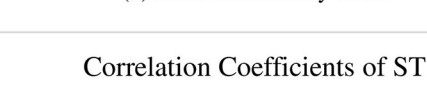
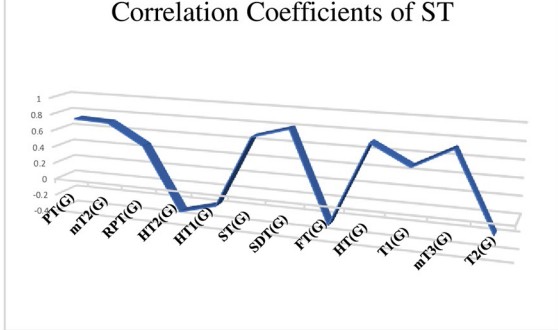

(f) Surface Tension on TI

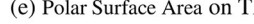
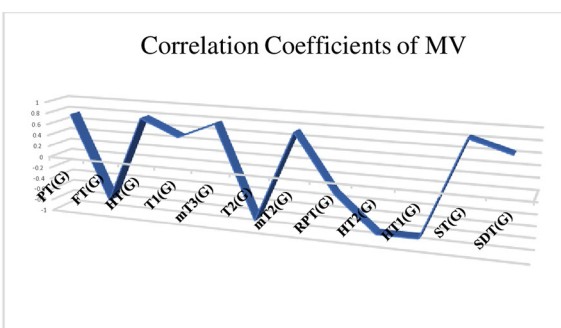

(g) Molar Volume on TI

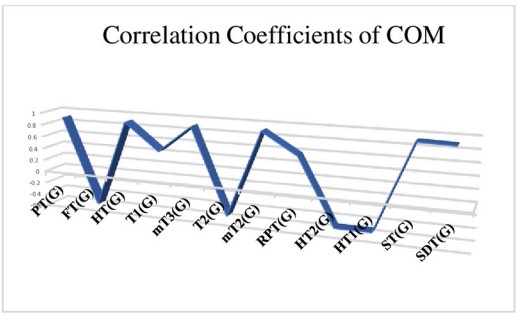

(h) Complexity on TI

**Fig 4. Physicochemical properties with topological indices.** https://doi.org/10.6084/m9.figshare.26988649.v1.

MR = 120.232–3735.217 [$T_2$ (G)], MV = 323.901–10116.120 [$T_2$ (G)]

**10. Sum connectivity temperature index [ST (G)]**

BP = 327.804+4.642 [ST (G)], EN = 54.435+.658 [ST (G)], FP = 149.323+2.682 [ST (G)]

MR = 42.390+.911 [ST (G)], PSA = 11.502+1.565 [ST (G)], ST = 37.512+.481 [ST (G)]

MV = 125.946+2.239 [ST (G)], COM = 127.395+7.726 [ST (G)]

## 4 Machine learning models for predictive analysis

In this study, machine learning models were employed to predict the physicochemical properties of drugs used in the treatment of Cancer. The primary goal was to assess the potential of these models in identifying complex and nonlinear relationships between molecular structures and physicochemical properties. The use of machine learning methods in drug analysis offers the advantage of uncovering hidden patterns within the data that traditional methods may fail to identify (Vapnik, 1995).

### 4.1. Rationale for using machine learning models

Machine learning models are particularly suitable for capturing intricate, nonlinear relationships in large datasets. This is crucial for predicting drug properties, as these relationships are not always straightforward or linear. In this study, machine learning models were used to model these complex patterns and predict key physicochemical properties of drugs. These properties are vital for drug design, as they influence the drug's behavior, efficacy, and safety profile. Traditional statistical methods often fail to account for these complexities, making machine learning an ideal choice.

For this analysis, in addition to linear regression, two other machine learning methods were used, which are described below:

1. **Support Vector Regression (SVR):** This model is well-known for its effectiveness in handling nonlinear data.

2. **Random Forest:** A model based on an ensemble of decision trees, which aggregates the predictions of many trees to improve accuracy and reduce overfitting. Random Forest is particularly effective for regression tasks in complex datasets.

These models were employed to predict the following physicochemical properties of the drugs: BP, EN, FP, MR, PSA, ST, MV, COM.

### 4.2. Comparison of prediction and analysis of models

Linear regression performed the best, effectively capturing the relationships between the molecular structure of the drugs and their physicochemical properties. The SVR model also captured complex patterns but showed weaker results compared to linear regression. Random Forest performed the least well among the models. Tables 14–17 illustrate the predictions of physicochemical properties using different models, and the evaluation results are presented in Table 17 below and Fig 5.

The linear regression model performed well in predicting most physical and chemical properties such as BP, EN, MR, and MV, with its predictions closely matching the actual values. Overall, the model is effective in modeling linear relationships.

The SVR model performed relatively well in predicting most physical and chemical properties, with predictions for BP**,** EN**,** MR, and MV being close to the actual values. Although the model showed reasonable accuracy for most properties, there were some discrepancies, especially for COM**.** Overall, the SVR model was effective in capturing complex, non-linear

**Table 14. Prediction of physical and chemical properties using linear regression.**

| Drugs | BP | EN | FP | MR | PSA | ST | MV | COM |
|---|---|---|---|---|---|---|---|---|
| Aminopterin | 502.9178 | 79.25277 | 262.1571 | 74.6904 | 62.24942 | 53.8369 | 174.9829 | 358.1457 |
| Convolutamide A | 594.2665 | 92.34224 | 266.1994 | 104.7399 | 99.77667 | 60.24787 | 276.7021 | 507.8311 |
| Convolutamydine A | 518.2478 | 81.03754 | 248.6585 | 67.98445 | 62.77917 | 54.50266 | 193.9989 | 370.6587 |
| Daunorubicin | 637.7867 | 98.20174 | 296.6422 | 129.8791 | 67.19081 | 52.72902 | 388.7722 | 79.93658 |
| Minocycline | 790.5111 | 117.5007 | 433.1623 | 112.5844 | 71.14037 | 68.90717 | 286.986 | 557.8229 |
| Podophyllotoxin | 813.7398 | 120.9429 | 420.0798 | 119.6955 | 147.3117 | 90.67948 | 298.7468 | 943.1044 |
| Caulibugulone E | 434.6103 | 67.74868 | 227.1406 | 61.39677 | 89.57761 | 62.5712 | 159.6744 | 642.1132 |
| PerfragilinA | 761.5278 | 118.4367 | 439.903 | 127.4511 | 197.2214 | 87.75863 | 344.8008 | 963.783 |
| Melatonin | 368.4921 | 63.03676 | 164.3571 | 49.57837 | 75.75286 | 50.06707 | 164.2359 | 188.6043 |
| Tambjamine k | 580.3727 | 90.93451 | 274.7751 | 85.99571 | 75.33736 | 58.83437 | 232.2445 | 390.0584 |
| Amathaspiramide E | 360.9571 | 71.81093 | 182.5549 | 72.85438 | 83.39079 | 45.07873 | 227.6464 | 104.2794 |
| Aspidostomide E | -158.097 | 30.76467 | 134.8301 | 54.24999 | 234.2071 | 58.70372 | 308.8265 | 919.6087 |

https://doi.org/10.6084/m9.figshare.28077872

relationships in the data, but linear regression performed better in providing more accurate predictions.

The Random Forest model showed acceptable results in predicting the physical and chemical properties of the drugs, but compared to the Linear Regression and SVR models, its accuracy was lower in some predictions. For instance, for properties like BP and EN, there were notable discrepancies between the predicted values and the actual values, indicating lower precision in these cases. Therefore, it can be concluded that the Linear Regression and SVR models performed better in most cases, with their predictions being closer to the actual values.

As depicted in Fig 5, Linear regression demonstrated the best performance overall. Random Forest excelled in predicting non-linear relationships in some cases but showed lower accuracy in others. The SVR model exhibited weak performance.

Based on the evaluation of machine learning models using the coefficient of determination ($R^2$) for predicting the physicochemical properties of drugs, linear regression demonstrated the best performance, achieving the highest $R^2$ values for most properties such as BP (0.95),

**Table 15. Prediction of physical and chemical properties using SVR.**

| Drugs | BP | EN | FP | MR | PSA | ST | MV | COM |
|---|---|---|---|---|---|---|---|---|
| Aminopterin | 594.9537 | 89.85808 | 264 | 99.86427 | 76.13678 | 59.12679 | 257.6979 | 542.093 |
| Convolutamide A | 597.9 | 93.6 | 266.3524 | 104.3 | 77.69741 | 59.49674 | 263.0683 | 543.2356 |
| Convolutamydine A | 595.1516 | 89.95788 | 264.1292 | 99.9392 | 76.19004 | 59 | 257.831 | 542.0788 |
| Daunorubicin | 600.0783 | 95.1467 | 267.7187 | 104.8855 | 78 | 59.11968 | 263.5021 | 542.3162 |
| Minocycline | 600.6772 | 95.18554 | 268.4679 | 105.0437 | 77.56055 | 61.31214 | 262 | 544.5428 |
| Podophyllotoxin | 601.0291 | 95.79631 | 269.0508 | 105.6726 | 79.49784 | 62.45004 | 263.8814 | 545.5563 |
| Caulibugulone E | 594.9814 | 89.89589 | 263.7808 | 99.74355 | 76.94799 | 59.89127 | 257.9344 | 543 |
| PerfragilinA | 600.6081 | 95.10032 | 268.7645 | 105.7717 | 79.61789 | 62.34736 | 263.6153 | 545.4268 |
| Melatonin | 595.0813 | 89.90094 | 264.2213 | 100.0452 | 76.27914 | 58.60532 | 258.1575 | 541.581 |
| Tambjamine k | 597.4321 | 92.19833 | 265.4958 | 102.2551 | 76.70804 | 59.64387 | 260.2539 | 542.5644 |
| Amathaspiramide E | 596.6249 | 91.67012 | 265.0912 | 102.1404 | 76.90827 | 58.59904 | 261.6032 | 541.9419 |
| Aspidostomide E | 598.0908 | 93.07143 | 266.3837 | 102.6357 | 78.01914 | 60.49908 | 263.9434 | 543.6364 |

https://figshare.com/articles/figure/_/28077872

**Table 16. Prediction of physical and chemical properties using random forest.**

| Drugs | BP | EN | FP | MR | PSA | ST | MV | COM |
|---|---|---|---|---|---|---|---|---|
| Aminopterin | 498.258 | 78.071 | 245.284 | 66.457 | 61.03 | 52.371 | 187.585 | 352.7 |
| Convolutamide A | 605.34 | 96.981 | 275.488 | 108.319 | 82.47 | 57.973 | 303.846 | 433.21 |
| Convolutamydine A | 510.428 | 80.512 | 253.288 | 68.335 | 65.53 | 57.259 | 188.059 | 368.43 |
| Daunorubicin | 657.242 | 102.172 | 339.641 | 122.584 | 85.62 | 53.639 | 363.324 | 182.19 |
| Minocycline | 779.287 | 113.26 | 395.533 | 111.29 | 76.27 | 70.345 | 265.033 | 505.38 |
| Podophyllotoxin | 783.54 | 118.527 | 429.158 | 116.84 | 144.64 | 84.586 | 306.266 | 880.7 |
| Caulibugulone E | 436.613 | 70.488 | 222.204 | 62.825 | 94.06 | 66.31 | 171.933 | 527.79 |
| PerfragilinA | 760.978 | 115.907 | 415.27 | 125.081 | 164.51 | 81.319 | 339.038 | 860.44 |
| Melatonin | 402.455 | 65.285 | 196.446 | 59.96 | 69.09 | 53.67 | 155.673 | 316.86 |
| Tambjamine k | 614.766 | 91.909 | 269.536 | 104.316 | 70.37 | 59.516 | 230.103 | 387.24 |
| Amathaspiramide E | 515.395 | 82.127 | 249.194 | 92.626 | 79.59 | 52.602 | 250.153 | 297.37 |
| Aspidostomide E | 641.264 | 104.75 | 330.928 | 96.938 | 114.34 | 68.999 | 314.702 | 707.87 |

https://figshare.com/articles/figure/_/28078031

EN (0.91), and MV (0.93), indicating strong predictive accuracy. Random Forest provided valuable insights into complex, non-linear relationships, though its accuracy was slightly lower than that of linear regression. Finally, SVR performed poorly and provided less accurate results compared to the other two models. Therefore, linear regression can be considered the best model for predicting the physicochemical properties of drugs.

## 5 Conclusion

Table 6 and Fig 3 illustrate the correlation between the physical and chemical properties of anti-cancer drugs and the defined temperature indices.

1. The Polar Surface Area is best predicted by the modified second temperature index, with a correlation coefficient (r) of 0.808.

2. The Sum Connectivity temperature index is the most effective predictor for Boiling Point (r = 0.836) and Molar Volume (r = 0.848). It also exhibits the highest significant correlations with Molar Refractivity (r = 0.924) and Complexity (r = 0.921).

3. The Symmetric Division temperature index shows a positive correlation with Enthalpy of Vaporization (r = 0.854), Flash Point (r = 0.857), and Surface Tension (r = 0.808).

This analysis reveals a positive correlation between the physical and chemical properties of Cancer drugs and the temperature indices. Tables 7–13 and 18–20 present regression models for various physical and chemical properties. The results demonstrate that the regression coefficients (r) exceed 0.6, and the p-values are below 0.05, indicating that these predictors are

**Table 17. Evaluation of advanced machine learning models based on the coefficient of determination ($R^2$) for predicting physicochemical properties of drugs.**

| Model | BP | EN | FP | MR | PSA | ST | MV | COM |
|---|---|---|---|---|---|---|---|---|
| **SVR** | -0.08055 | -0.02977 | 0.017391 | 0.046939 | -0.14016 | 0.041684 | 0.046667 | -0.01107 |
| **Random Forest** | 0.224961 | 0.064874 | 0.322626 | 0.885989 | 0.630048 | 0.638303 | 0.946325 | 0.892501 |
| **Linear Regression** | **0.952123** | **0.91647** | 0.860965 | 0.558735 | **0.923733** | 0.482884 | **0.939592** | 0.827118 |

https://figshare.com/articles/figure/_/28078031

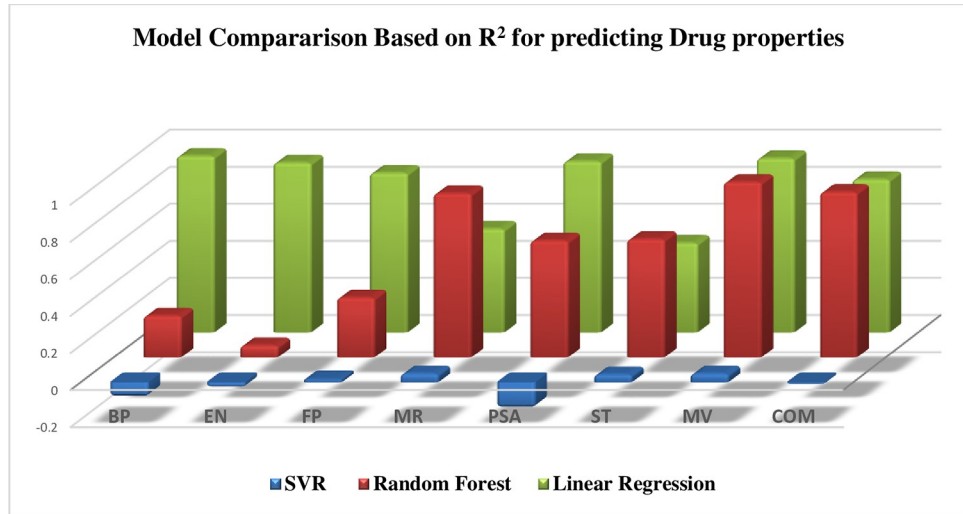

**Fig 5. Comparison of machine learning models for predicting physicochemical properties of cancer drugs.** https://figshare.com/articles/figure/_/28078031.

**Table 18. Statistical metrics for the linear QSPR model applied to $HT_1$ (G).**

| Physical Properties | N | A | b | R | $R^2$ | SE | F | P | Indicator |
|---|---|---|---|---|---|---|---|---|---|
| MR | 12 | -39.078 | 146.930 | .772 | .597 | 14.753 | 14.788 | .000 | significant |
| MV | 12 | -116.486 | 410.627 | **.859** | **.737** | 31.925 | 28.061 | .000 | significant |

The significance of bold numbers denote highest correlation value

https://doi.org/10.6084/m9.figshare.26997166.v1

**Table 19. Statistical metrics for the linear QSPR model applied to $T_2$ (G).**

| Physical Properties | N | A | b | R | $R^2$ | SE | F | P | Indicator |
|---|---|---|---|---|---|---|---|---|---|
| MR | 12 | -167.663 | 146.781 | .758 | .575 | 15.364 | 13.503 | .000 | significant |
| MV | 12 | -503.865 | 411.470 | **.850** | **.722** | 33.321 | 25.928 | .000 | significant |

The significance of bold numbers denote highest correlation value

https://doi.org/10.6084/m9.figshare.26997214.v1

**Table 20. Statistical metrics for the linear QSPR model applied to ST (G).**

| Physical Properties | N | A | b | R | $R^2$ | SE | F | P | Indicator |
|---|---|---|---|---|---|---|---|---|---|
| BP | 11 | 4.642 | 327.804 | .836 | .700 | 61.623 | 20.968 | .000 | significant |
| EN | 11 | .658 | 54.435 | **.860** | **.740** | 7.893 | 25.674 | .000 | significant |
| FP | 11 | 2.682 | 149.323 | .767 | .588 | 45.519 | 12.830 | .010 | significant |
| MR | 12 | .911 | 42.390 | .924 | .854 | 7.471 | 58.536 | .000 | significant |
| PSA | 12 | 1.565 | 11.502 | .764 | .584 | 26.243 | 14.014 | .004 | significant |
| ST | 12 | .481 | 37.512 | .682 | .464 | 10.263 | 8.673 | .004 | significant |
| MV | 12 | 2.239 | 125.946 | .848 | .718 | 27.831 | 25.508 | .001 | significant |
| COM | 11 | 7.726 | 127.395 | .921 | .848 | 67.000 | 50.092 | .000 | significant |

The significance of bold numbers denote highest correlation value

https://doi.org/10.6084/m9.figshare.26997193.v1

reliable for linear regression. The equations are formulated based on criteria such as minimum standard error (SE), maximum R-squared ($R^2$), and maximum F-statistic. Consequently, it can be concluded that all physical and chemical properties are highly significant. This underscores the potential value of these topological indices in QSPR analysis for Cancer drugs, as evidenced by the plotted regression lines. The study's findings can be applied to the production, development, and enhancement of more effective Cancer drugs. The theoretical insights derived from this study are beneficial for the development of new cancer therapies. Our findings reveal a clear trend in examining drug structures and their physical characteristics. Ultimately, this research contributes to the efficient design of new drugs and the development of preventive measures for the diseases in question. The principles of QSPR and topological indices offer valuable new approaches for estimating properties related to specific diseases and drugs, as demonstrated by the conclusions of this study. Furthermore, when comparing the three methods, despite the simplicity of Linear Regression, it consistently showed the best performance in predicting the physical and chemical properties of cancer drugs, outperforming both the SVR and Random Forest models. This emphasizes the effectiveness of Linear Regression in capturing the relationships within the data.

## Author Contributions

**Conceptualization:** Xiaolong Shi, Saeed Kosari, Masoud Ghods, Negar Kheirkhahan.

**Data curation:** Masoud Ghods.

**Formal analysis:** Xiaolong Shi, Negar Kheirkhahan.

**Funding acquisition:** Xiaolong Shi.

**Investigation:** Saeed Kosari, Masoud Ghods.

**Methodology:** Saeed Kosari.

**Project administration:** Xiaolong Shi.

**Resources:** Negar Kheirkhahan.

**Software:** Saeed Kosari.

**Supervision:** Xiaolong Shi, Saeed Kosari.

**Validation:** Xiaolong Shi, Negar Kheirkhahan.

**Writing – original draft:** Saeed Kosari, Masoud Ghods.

**Writing – review & editing:** Masoud Ghods, Negar Kheirkhahan.

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
