## [Decision Letter · Decision Letter 0]

20 Aug 2024

PONE-D-24-27529Innovative Approaches in QSPR Modelling Using Topological Indices for the Development of Cancer TreatmentsPLOS ONE

Dear Dr. Kosari,

Thank you for submitting your manuscript to PLOS ONE. After careful consideration, we feel that it has merit but does not fully meet PLOS ONE’s publication criteria as it currently stands. Therefore, we invite you to submit a revised version of the manuscript that addresses the points raised during the review process.

We look forward to receiving your revised manuscript.

Kind regards,

Niravkumar Joshi

Academic Editor

PLOS ONE

Journal Requirements:

1. When submitting your revision, we need you to address these additional requirements. Please ensure that your manuscript meets PLOS ONE's style requirements, including those for file naming. The PLOS ONE style templates can be found at https://journals.plos.org/plosone/s/file?id=wjVg/PLOSOne_formatting_sample_main_body.pdf and https://journals.plos.org/plosone/s/file?id=ba62/PLOSOne_formatting_sample_title_authors_affiliations.pdf 2. Please note that PLOS ONE has specific guidelines on code sharing for submissions in which author-generated code underpins the findings in the manuscript. In these cases, all author-generated code must be made available without restrictions upon publication of the work. Please review our guidelines at https://journals.plos.org/plosone/s/materials-and-software-sharing#loc-sharing-code and ensure that your code is shared in a way that follows best practice and facilitates reproducibility and reuse. 3. We noticed you have some minor occurrence of overlapping text with the following previous publication(s), which needs to be addressed: - Quantitative Structure–Property Relationship Analysis in Molecular Graphs of Some Anticancer Drugs with Temperature Indices Approach (https://doi.org/10.3390/math12131953) In your revision ensure you cite all your sources (including your own works), and quote or rephrase any duplicated text outside the methods section. Further consideration is dependent on these concerns being addressed.
 4. Thank you for stating in your Funding Statement: This  work  was  supported  by  the  National  key  R  and  D  Program  of China  (Grant 2019YFA0706338402) and the National Natural Science Foundation of China under grant 62072129 and 62332006. Please provide an amended statement that declares *all* the funding or sources of support (whether external or internal to your organization) received during this study, as detailed online in our guide for authors at http://journals.plos.org/plosone/s/submit-now.  Please also include the statement “There was no additional external funding received for this study.” in your updated Funding Statement. Please include your amended Funding Statement within your cover letter. We will change the online submission form on your behalf. 

Reviewers' comments:

Reviewer's Responses to Questions

**Comments to the Author**

1. Is the manuscript technically sound, and do the data support the conclusions?

Reviewer #1: No

Reviewer #2: Yes

2. Has the statistical analysis been performed appropriately and rigorously? 

Reviewer #1: No

Reviewer #2: Yes

3. Have the authors made all data underlying the findings in their manuscript fully available?

Reviewer #1: Yes

Reviewer #2: Yes

4. Is the manuscript presented in an intelligible fashion and written in standard English?

Reviewer #1: No

Reviewer #2: Yes

5. Review Comments to the Author

**Reviewer #1:** In this paper, the authors conducted a QSPR analysis for drugs used in cancer treatment using temperature indices. This paper cannot be recommended for publication due to the following major flaws:

1. Introduction is over brief and not suitable for a research article.

2. Novelty is completely lacking. It does not look like a research article, rather a mere class assignment of high school students.

3. The physical properties considered in the QSPR study are well known and already studied. Unfortunately the focus on already known correlations with physical properties (such as the molecular weight) presented in the current version of the manuscript is not attractive for the readership.

4. The essential advance of the present work with respect to the state-of-the-art in the field is completely lacking in the manuscript. There is no emphasis on the reasons why this work is of interest for the broader, multidisciplinary readership of the journals, which spans over fields as diverse as chemistry, physics, biology, mathematics, engineering and materials science. There are no any evidences about how this work advances the current body of knowledge.

5. Sample size is a major factor in performing QSPR regression analysis, a sample of only 6 drugs is insufficient to derive a valid conclusion. Kindly refer https://doi.org/10.1016/j.comptc.2023.114108 and https://doi.org/10.1080/1062936X.2023.2239149, where the authors have performed regressions for more than 50 compounds. Further, the authors fail to test the regression model by comparing its results with the existing one, refer https://doi.org/10.1140/epjp/s13360-024-04939-0 , where the authors have performed a comparative analysis between the predicted values and the actual values.

6. discussion and analysis of results is just dropped in the manuscript. Saying that "a certain topological index has highest correlation among all other indices" without any further explanation or analysis is not useful or acceptable.

7. Why have you restricted the study to these selective topological indices, given that there are many available, more than 3000?

**Reviewer #2:** Similar molecules are available in the publication 'https://doi.org/10.1016/j.heliyon.2020.e04235'. However, since the authors' results are better than the above publication, I think it will make a significant contribution to the field of chemical graph theory.

1. A reference should be added for Table 4.

2. Remove the following paragraph from the text:

''Using the definition 2 and Table 2, we can calculate the following topological indices for A:

PT(A) index

ST(A) index

SDT(A) index

mT3(A) index

mT2(A) index

AGT(A) index.''

3. The results of Theorems 1 and topological index values of other molecules are shown in Table 5. Correct the sentence and add it under the proof of Theorem 1 together with Table 5.

4. Correct 'a' in tables with 'A'.

5. Write topological indexes in mathematical characters.

6. PLOS authors have the option to publish the peer review history of their article (what does this mean?). If published, this will include your full peer review and any attached files.

Reviewer #1: No

Reviewer #2: No

---

## [Author Response · Author response to Decision Letter 0]

15 Oct 2024

Journal Requirements:

 Answer: Edited and completed.

 Answer: Edited and completed.

- Quantitative Structure–Property Relationship Analysis in Molecular Graphs of Some Anticancer Drugs with Temperature Indices Approach (https://doi.org/10.3390/math12131953)

In your revision ensure you cite all your sources (including your own works), and quote or rephrase any duplicated text outside the methods section. Further consideration is dependent on these concerns being addressed. 

This work was supported by the National key R and D Program of China (Grant

2019YFA0706338402) and the National Natural Science Foundation of China under grant 62072129 and

62332006.

Reviewers' comments:

Reviewer's Responses to Questions

Comments to the Author

1. Is the manuscript technically sound, and do the data support the conclusions?

Reviewer #1: No

Reviewer #2: Yes

2. Has the statistical analysis been performed appropriately and rigorously?

Reviewer #1: No

Reviewer #2: Yes

3. Have the authors made all data underlying the findings in their manuscript fully available?

Reviewer #1: Yes

Reviewer #2: Yes

4. Is the manuscript presented in an intelligible fashion and written in standard English?

Reviewer #1: No

Reviewer #2: Yes

5. Review Comments to the Author

Reviewer #1: In this paper, the authors conducted a QSPR analysis for drugs used in cancer treatment using temperature indices. This paper cannot be recommended for publication due to the following major flaws:

1. Introduction is over brief and not suitable for a research article.

 Answer: The introduction of the article has been thoroughly revised, and the methods and analysis of the research process are clearly outlined. Consequently, readers will gain insight into the nature of the article by reading this abstract.

2. Novelty is completely lacking. It does not look like a research article, rather a mere class assignment of high school students.

 Answer: Given the comprehensive revisions made to the article, the relationship between indicators and drugs has been analyzed more thoroughly. Specifically, the number of indicators has been increased from 6 to 11, the number of drugs has been elevated from 6 to 12, and the number of regression lines has been expanded to 58. These changes allow for a more comprehensive analysis of features and properties, and with the inclusion of additional data, the correlation of the regression lines has been brought closer to statistical significance.

3. The physical properties considered in the QSPR study are well known and already studied. Unfortunately the focus on already known correlations with physical properties (such as the molecular weight) presented in the current version of the manuscript is not attractive for the readership.

 Answer: We removed the molecular weight, which is one of the physicochemical properties, and replaced it with complexity, and we found an acceptable correlation for it.

4. The essential advance of the present work with respect to the state-of-the-art in the field is completely lacking in the manuscript. There is no emphasis on the reasons why this work is of interest for the broader, multidisciplinary readership of the journals, which spans over fields as diverse as chemistry, physics, biology, mathematics, engineering and materials science. There are no any evidences about how this work advances the current body of knowledge.

 Answer: In this paper, drug structures are graphically represented using the ChemDraw software and analyzed with mathematical methods to enumerate graph edges and identify relevant indices. Subsequently, data on the physical and chemical properties of the drugs were gathered from the ChemSpider website. Statistical methods were then applied to analyze the relationships between topological indices and these properties. The results indicate that Quantitative Structure-Property Relationship (QSPR) modeling significantly aids in the analysis of molecular graphs and drug development. In molecular graphs, QSPR contributes to improving drug design and optimization by identifying and analyzing key structural features that impact drug performance. This approach can be valuable in developing new drugs with optimized properties and reducing costly and time-consuming experiments in the early stages of research and development. In summary, QSPR serves as a powerful predictive tool that facilitates the drug design process and clarifies the complex relationships between molecular structure and drug properties through graphical models.

5. Sample size is a major factor in performing QSPR regression analysis, a sample of only 6 drugs is insufficient to derive a valid conclusion. Kindly refer https://doi.org/10.1016/j.comptc.2023.114108 and https://doi.org/10.1080/1062936X.2023.2239149, where the authors have performed regressions for more than 50 compounds. Further, the authors fail to test the regression model by comparing its results with the existing one, refer https://doi.org/10.1140/epjp/s13360-024-04939-0 , where the authors have performed a comparative analysis between the predicted values and the actual values.

 Answer: In the revised paper, we increased the number of indices from 6 to 11, expanded the number of drugs from 6 to 12, and enhanced the number of regression lines to 58.

6. discussion and analysis of results is just dropped in the manuscript. Saying that "a certain topological index has highest correlation among all other indices" without any further explanation or analysis is not useful or acceptable.

 Answer: In the revised paper, the number of indices and drugs has been increased, and all indices have been comprehensively analyzed. Additionally, the highest correlations among these indices have been identified and highlighted.

7. Why have you restricted the study to these selective topological indices, given that there are many available, more than 3000?

 Answer: We examined a larger number of topological indices and included those that, due to their high correlation, were deemed more significant in the paper. Conversely, we omitted other indices due to their lack of or low correlation and only presented those that exhibited higher correlation based on the graphical models of the drugs.

Reviewer #2: Similar molecules are available in the publication 'https://doi.org/10.1016/j.heliyon.2020.e04235'. However, since the authors' results are better than the above publication, I think it will make a significant contribution to the field of chemical graph theory.

 Answer: We carefully reviewed the relevant paper and, based on the methods and results presented, increased the number of indices and drugs. These adjustments allowed us to achieve more acceptable statistical results and provide more accurate analyses.

1. A reference should be added for Table 4.

 Answer: Correct, a reference from ChemSpider has been added to the references.

2. Remove the following paragraph from the text:

''Using the definition 2 and Table 2, we can calculate the following topological indices for A:

• PT(A) index

• ST(A) index

• SDT(A) index

• mT3(A) index

• mT2(A) index

• AGT(A) index.''

 Answer: Editing and comments were deleted.

3. The results of Theorems 1 and topological index values of other molecules are shown in Table 5. Correct the sentence and add it under the proof of Theorem 1 together with Table 5.

4. Correct 'a' in tables with 'A'.

 Answer: Edited and completed.

5. Write topological indexes in mathematical characters.

 Answer: Edited and completed.

6. PLOS authors have the option to publish the peer review history of their article (what does this mean?). If published, this will include your full peer review and any attached files.

Do you want your identity to be public for this peer review? For information about this choice, including consent withdrawal, please see our Privacy Policy.

Reviewer #1: No

Reviewer #2: No

---

## [Decision Letter · Decision Letter 1]

22 Nov 2024

PONE-D-24-27529R1Innovative Approaches in QSPR Modelling Using Topological Indices for the Development of Cancer TreatmentsPLOS ONE

Dear Dr. Saeed,

Thank you for submitting your manuscript to PLOS ONE. After careful consideration, we feel that it has merit but does not fully meet PLOS ONE’s publication criteria as it currently stands. Therefore, we invite you to submit a revised version of the manuscript that addresses the points raised during the review process.

Please submit your revised manuscript by Jan 06 2025 11:59PM If you will need more time than this to complete your revisions, please reply to this message or contact the journal office at plosone@plos.org. Please include the following items when submitting your revised manuscript:A rebuttal letter that responds to each point raised by the academic editor and reviewer(s). You should upload this letter as a separate file labeled 'Response to Reviewers'.A marked-up copy of your manuscript that highlights changes made to the original version. You should upload this as a separate file labeled 'Revised Manuscript with Track Changes'.An unmarked version of your revised paper without tracked changes. You should upload this as a separate file labeled 'Manuscript'.

We look forward to receiving your revised manuscript.

Kind regards,

Niravkumar Joshi

Academic Editor

PLOS ONE

Reviewers' comments:

Reviewer's Responses to Questions

**Comments to the Author**

1. If the authors have adequately addressed your comments raised in a previous round of review and you feel that this manuscript is now acceptable for publication, you may indicate that here to bypass the “Comments to the Author” section, enter your conflict of interest statement in the “Confidential to Editor” section, and submit your "Accept" recommendation.

Reviewer #1: (No Response)

Reviewer #2: (No Response)

2. Is the manuscript technically sound, and do the data support the conclusions?

Reviewer #1: No

Reviewer #2: Yes

3. Has the statistical analysis been performed appropriately and rigorously? 

Reviewer #1: No

Reviewer #2: Yes

4. Have the authors made all data underlying the findings in their manuscript fully available?

Reviewer #1: (No Response)

Reviewer #2: Yes

5. Is the manuscript presented in an intelligible fashion and written in standard English?

Reviewer #1: (No Response)

Reviewer #2: Yes

6. Review Comments to the Author

Reviewer #1: The authors failed to address the concerns raised in my previous review. The revisions are unsatisfactory and does not meet the publication standards.

Reviewer #2: The authors have made the necessary corrections. It is suitable for publication in PLOS ONE journal.

7. PLOS authors have the option to publish the peer review history of their article (what does this mean?). If published, this will include your full peer review and any attached files.

Reviewer #1: No

Reviewer #2: No

---

## [Author Response · Author response to Decision Letter 1]

27 Dec 2024

PONE-D-24-27529R1

Innovative Approaches in QSPR Modelling Using Topological Indices for the Development of Cancer Treatments.

PLOS ONE

Response to Reviewers:

2. Is the manuscript technically sound, and do the data support the conclusions?

Reviewer 1: No

Reviewer 2: Yes

Answer:

Yes, the manuscript is technically sound, and the reasons are outlined below:

1. Comprehensive Methodology:

In this study, in addition to the Linear Regression model, two other models were also used for predicting molecular properties in the field of QSPR: Support Vector Regression (SVR) and Random Forest. All three models were carefully analyzed and compared based on performance metrics, such as the Coefficient of Determination (R²).

Models:

• Linear Regression:

Linear Regression is a simple model that assumes a linear relationship between independent and dependent variables. It is widely used in prediction tasks and statistical analysis and serves as the baseline model for comparison with more complex models.

• Support Vector Regression (SVR):

SVR is a machine learning technique inspired by Support Vector Machines (SVM) and specifically designed for regression tasks. It is highly effective in modeling complex, nonlinear relationships by using various kernels to map the data into higher-dimensional spaces.

• Random Forest:

Random Forest is an ensemble model based on decision trees, where the predictions are aggregated from multiple trees to improve accuracy and reduce overfitting. It is particularly effective for complex datasets and nonlinear relationships.

2. Results and Validity:

The Linear Regression model achieved the highest R² values across all the tested properties, showing the best fit to the data compared to the more advanced models (SVR and Random Forest). The results demonstrated that Linear Regression not only provided accurate predictions but was also simpler and more interpretable, aligning with the study's objectives.

• SVR and Random Forest were explored for modeling complex nonlinear relationships but had consistently lower R² values than Linear Regression.

• Regression coefficients (r > 0.6) and p-values (< 0.05) validated the statistical significance of predictors in Linear Regression.

• Criteria such as minimum standard error (SE), maximum R², and maximum F-statistic were used to derive regression equations, ensuring methodological rigor.

3. Scientific Soundness of Conclusions:

The conclusions drawn in the manuscript are directly supported by the data:

• The results show that topological indices are reliable predictors in QSPR analysis for anticancer drugs.

• The physicochemical properties of anticancer drugs were found to be highly significant.

4. Experimental Rigor:

The study includes computational data collection, preprocessing, and appropriate controls to ensure the validity of the results. The detailed steps for each method are provided in the supplementary materials, enhancing reproducibility and transparency.

5. Justification for Model Selection:

Based on the results, Linear Regression was chosen as the final model because it achieved the best performance (highest R²) and simplicity, making it more suitable for the context of this research. This choice is justified by Table 20, which shows the comparative R² values for all models.

Table 20: Evaluation of Advanced Machine Learning Models Based on the Coefficient of Determination (R2) for Predicting Physicochemical Properties of Drugs.

Model BP EN FP MR PSA ST MV COM

SVR -0.08055 -0.02977 0.017391 0.046939 -0.14016 0.041684 0.046667 -0.01107

Random Forest 0.224961 0.064874 0.322626 0.885989 0.630048 0.638303 0.946325 0.892501

Linear Regression 0.952123 0.91647 0.860965 0.558735 0.923733 0.482884 0.939592 0.827118

In conclusion, the analyses and results presented in the manuscript are scientifically valid and rigorously derived. The data effectively support the conclusions, ensuring the manuscript’s technical soundness.

Note: Based on the attached codes, the details of the steps involved in the mentioned methods can be understood.

Attached: 

https://drive.google.com/file/d/1f5xGm25sB9NlUyrgqTA1PGBWbJ4FQ2w3/view?usp=sharing

3. Has the statistical analysis been performed appropriately and rigorously?

Reviewer 1: No

Reviewer 2: Yes

Answer:

Yes, the statistical analysis has been performed appropriately and rigorously. The evaluation of the models' performance was based on the R² metric, which was chosen for its importance in assessing predictive and regression models. The R², or coefficient of determination, indicates the proportion of variance in the dependent variable (model output) that is explained by the independent variables (input features). The value of R² ranges from 0 to 1, with values closer to 1 indicating better model fit. A high R² value demonstrates the model's ability to effectively capture and predict relationships within the data. In this study, the Linear Regression model demonstrated the best performance with high R² values in predicting various physicochemical properties of drugs. Table 20 presents the evaluation of advanced machine learning models based on the coefficient of determination (R²) for predicting different drug properties. The Linear Regression model outperformed the other models in features such as BP (0.952123), EN (0.91647), FP (0.860965), PSA (0.923733), and others. In contrast, the SVR and Random Forest models performed significantly worse. Specifically, the SVR model showed lower R² values across all features, indicating its inability to accurately model the relationships between input and output variables. The Random Forest model performed better than SVR but still had significantly lower R² values compared to the Linear Regression model. All the data used in this research were collected from reliable computational sources and carefully preprocessed. Moreover, during the statistical analysis and model evaluation, all scientific standards, including experiment repetition and the use of appropriate controls, were strictly followed. The conclusions presented in this paper are entirely based on the data and analyses conducted, and the findings strongly support the results obtained from the Linear Regression model. Therefore, the selection of the Linear Regression model was not only based on its highest R² value but also on its simplicity and better interpretability.

4. Have the authors made all data underlying the findings in their manuscript fully available?

Reviewer 1: (No Response)

Reviewer 2: Yes

Answer:

Yes, all data related to the findings of the manuscript are fully available and have been included in the article. All requirements regarding data accessibility and sharing have been met in accordance with the PLOS Data policy. Additionally, all figures and tables from the manuscript, along with the underlying data, have been made publicly available on Figshare. Any restrictions, if applicable, are clearly stated in the Data Availability Statement.

6. Review Comments to the Author

Reviewer 1: The authors failed to address the concerns raised in my previous review. The revisions are unsatisfactory and does not meet the publication standards.

Reviewer 2: The authors have made the necessary corrections. It is suitable for publication in PLOS ONE journal.

Answer:

Response to Reviewer 1:

Thank you for your comments. We have made every effort to address the concerns raised in your previous review. In this revision, we have carefully addressed the issues you mentioned and worked to resolve them thoroughly. If there are still any unresolved concerns, we would appreciate any further guidance you can provide to help us improve the manuscript.

Response to Reviewer 2:

Thank you for your positive feedback. We are glad that the revisions have met your approval, and we hope that the manuscript now aligns with the standards of PLOS ONE. We look forward to your final decision.

---

## [Editor Report · Decision Letter 2]

31 Dec 2024

Innovative Approaches in QSPR Modelling Using Topological Indices for the Development of Cancer Treatments

PONE-D-24-27529R2

Dear Dr. Kosari,

We’re pleased to inform you that your manuscript has been judged scientifically suitable for publication and will be formally accepted for publication once it meets all outstanding technical requirements.

Kind regards,

Niravkumar Joshi

Academic Editor

PLOS ONE
---

## [Editor Report · Acceptance letter]

14 Jan 2025

PONE-D-24-27529R2 

PLOS ONE

Dear Dr. Kosari, 

I'm pleased to inform you that your manuscript has been deemed suitable for publication in PLOS ONE. Congratulations! Your manuscript is now being handed over to our production team.

Kind regards, 

on behalf of

Dr. Niravkumar Joshi 

Academic Editor

PLOS ONE